# Chromatin state architecture governs transcription factor accessibility across plant genomes

Vikas Shukla[1,2], Elin Axelsson[1], Tetsuya Hisanaga[1,3], Jim Haseloff[4], Frédéric Berger[1]*, Facundo Romani[4]*

1 Gregor Mendel Institute, Austrian Academy of Sciences, BioCenter, Vienna, Austria, 2 Vienna BioCenter PhD Program, Doctoral School of the University of Vienna and Medical University of Vienna, Vienna, Austria, 3 Current address: Biological Sciences, Nara Institute of Science and Technology, Ikoma, Nara, Japan, 4 Department of Plant Sciences, University of Cambridge, Cambridge, United Kingdom

* frederic.berger@gmi.oeaw.ac.at (FB); fr391@cam.ac.uk (FR)

## Abstract

The complexity of varied modifications of chromatin composition is integrated in archetypal combinations called chromatin states that predict the local potential for transcription. The degree of conservation of chromatin states has not been established amongst plants, and how they interact with transcription factors is unknown. Here we identify and characterize chromatin states in the flowering plant *Arabidopsis thaliana* and the bryophyte *Marchantia polymorpha*, showing a large degree of functional conservation over more than 450 million years of land plant evolution. We used this new resource of conserved plant chromatin states to understand the influence of chromatin states on gene regulation. We established the preferential association of chromatin states with binding sites and activity of transcription factors. These associations define three main groups of transcription factors that bind upstream of the transcription start site, at the +1 nucleosome or further downstream of the transcription start site and broadly associate with distinct biological functions including a list of potential candidate pioneer factors we know little about in plants, compared to their important roles in animal stem cells and early development.

## Author summary

In eukaryotes, DNA is tightly associated with histone proteins. Histone covalent modifications and histones isoforms, also called histone variants provide most of the complexity of chromatin and associate in specific combinations called chromatin states. Here we establish the broad conservation of chromatin states and their association with transcription in land plants. We design an index that predicts transcription factor binding and identify distinct groups of transcription factors based on their occupation of chromatin states. Our findings suggest that chromatin defines a specific environment that reflects transcriptional activity.

**Data availability statement:** Datasets from the analyses are provided as supplementary data files. Pipeline and scripts used for data analysis are available on the following GitHub link: https://github.com/Gregor-Mendel-Institute/vikas_states_tf_study_2025.git. NGS data are deposited in GEO under accession numbers: GSE302232.

**Funding:** This project was funded by the Austrian academy of sciences with the following grants from the Austrian Research Fund to FB: FWF P32054, P36231, PAT1104523 and PAT6138924, by the European Union's Framework Programme for Research and Innovation Horizon 2020 (2014–2020) to TH: the Marie Curie Skłodowska Grant Agreement no. 847548 (VIP2), by the Leverhulme Trust: Early Career Fellowship (ECF-2023-534 to FR) and the Isaac Newton Trust (23.08(f) to FR). The funders had no role in study design, data collection and analysis, decision to publish, or preparation of the manuscript.

**Competing interests:** The authors have declared that no competing interests exist.

## Introduction

Eukaryotic genomes are organized into chromatin, a dynamic polymer of DNA and histone proteins that control access to the underlying genetic information. Chromatin structure, shaped by nucleosome positioning, histone modifications, histone variants, DNA methylation, and higher-order interactions, plays a fundamental role in gene regulation, genome stability, and chromosome organization. Advances in genome-wide profiling techniques have enabled the classification of chromatin states, defined by combinatorial patterns of histone marks and other epigenetic features, which reflect functionally distinct regulatory environments [1–3]. Chromatin states provide a powerful framework to annotate genome function, delineate regulatory elements, and uncover specialised domains of chromatin within constitutive heterochromatin occupied by transposable elements (TEs), facultative heterochromatin occupied by genes transcriptionally repressed and euchromatin occupied by expressed genes.

A major function of chromatin is to regulate transcription by controlling the accessibility of DNA to transcription factors (TFs). In turn, TFs bind to specific cis-regulatory elements and influence gene expression by recruiting transcriptional machinery and chromatin-modifying complexes. In metazoans, models of chromatin states have proven especially useful in distinguishing functional modes of TF activity [4–6]. Some TFs preferentially bind to open, transcriptionally active chromatin, while others engage nucleosomal DNA in repressive contexts, functioning as pioneer TFs that remodel chromatin to allow binding of additional TFs [6–9]. These distinct modes of chromatin engagement are often predictive of TF mechanisms, such as the recruitment of the Polycomb repressive complex (PRC) [10–12] or histone acetyltransferases (HATs) [13,14].

In plants, chromatin state analysis has primarily focused on *Arabidopsis thaliana*, where recent efforts have defined high-resolution maps covering developmental stages and environmental conditions [15,16]. This architecture shares analogous features with chromatin states in animals. These analyses constitute a valuable resource for functional genomics, but outside *Arabidopsis thaliana* reports of chromatin states are scarce [17,18] and the extent to which these combinatorial patterns of histone modifications are maintained across plant evolution remains an open question.

In contrast to animals, the interplay between TFs and chromatin has remained largely unexplored in plants. Early studies in *Arabidopsis thaliana* reported that most TFs bind open chromatin regions, with little variation in chromatin state preference across TF families [19], and the functional association between chromatin and TFs has remained unresolved. Notably, the best characterized pioneer TF so far is LEAFY (LFY), but other candidates among the MADS-box transcription factors have also been proposed as potential pioneer factors [20–22]. Although hundreds of TF binding profiles have been generated in *Arabidopsis*, the field lacks a unified framework to relate these profiles to chromatin context and transcriptional activity. Genome-wide TF binding assays such as ChIP-seq or DAP-seq often reveal that many binding events are non-productive, i.e., they do not lead to measurable changes in gene expression [23,24]. Unlike in animals, there is no unified chromatin–TF framework akin to ENCODE or modENCODE [25], complicating efforts to classify TFs based on binding context or regulatory output.

Here, by comparing *Arabidopsis thaliana* and *Marchantia polymorpha,* we showed the overall conservation of the role of histone H2A variants and histone post-translational modifications in defining conserved chromatin states. To characterize TF association with chromatin states, we defined a TF activity score aggregating both genome-wide TF binding data and co-regulation of gene expression. Using this strategy, we showed that TFs can be grouped into distinct categories based on chromatin state preferences. These are involved in distinct biological functions, including a subset of potential candidate pioneer factors. We propose that TF–chromatin relationships follow similar principles in both species, supporting the conservation of regulatory strategies across land plants.

## Results

### Chromatin states exhibit positional preference for gene regulatory regions of *Arabidopsis thaliana*

Chromatin states integrate the specific combinatorial occupancy of histone modifications and variants [2] at a near nucleosome resolution (bin size = 200 bp; see Methods). These states help delineate major functional domains of chromatin and provide insights into transcriptional regulation. We independently used ChromHMM [2] to recalculate the chromatin states of the *Arabidopsis thaliana* genome (TAIR 10) using the series of genomic profiles of histone variants and most abundant histone PTMs used by [15]. We fine-tuned the chromatin state annotation pipeline and used stricter quality control steps (see Methods), to obtain a refined and more robust definition of chromatin states [15,16]. Each group of states were defined by a specific combination of histone variants and histone PTMs (Fig 1A) and were associated with diverse levels of transcripts (S1A Fig), DNA methylation (S1B–S1D Fig), and were differentially associated with transposable elements, genes or intergenic regions (Fig 1B).

The new model contained 26 states including seven states of constitutive heterochromatin (H1 − H7) on transposable elements, four intergenic states (I1 − I4), five states of facultative heterochromatin (F1 − F5) on lowly expressed protein coding genes, and ten states in euchromatin on expressed genes (E1 − E10) (Fig 1A). The enrichment in specific classes of H2A variants marked the identity of each group of states, with H2A.W in constitutive heterochromatin, H2A.Z in facultative heterochromatin, and H2A (and H2A.X) on expressed genes. Only minor differences separated this set of chromatin states from those previously described [15]. The previous states H4 becomes annotated as H2, and the previous states H2 and H3 split in states H3 − H5, and the reannotation of the facultative chromatin states including the state F1 split into I1 and I2 (S1E Fig).

We investigated the positional preference of chromatin states along chromosomes and observed that the states of constitutive heterochromatin (H1-H7) were primarily enriched on transposable elements and showed a positional preference with respect to the centromere of chromosomes (S1F Fig). The state H1 showed an exclusive preference for centromeric chromatin while states H2 to H7 were successively enriched from pericentromeres (H2-H5) to chromosome arms (H6 and H7) (S1F Fig). This constitutive heterochromatin represents circa 20% of the genome (S1G Fig).

The other states were primarily enriched at genes, and we performed neighbourhood enrichment analysis with the Transcription Start Site (TSS) or the Transcription Termination Site (TTS) as anchors. (Fig 1C; see Methods). In euchromatin, expressed genes were occupied by a remarkable sequential enrichment pattern of chromatin states. The state I4 was most specifically enriched in the promoter and terminator regions of most protein-coding genes while the states E1 - E8 were organized from the promoter towards the gene body. The states E1, E2, and E3 showed enrichment in positions corresponding approximately to the 1$^{st}$, 2$^{nd}$, and 3$^{rd}$ nucleosomes, respectively, for most of the expressed protein-coding genes. The states E4 and E5 occupied gene bodies and E5 was also marked by high CG methylation and enrichment in H3.3 compared with state E4 (Figs 1A, 1C and S1B), states E6, E7, and E8 occupied the end of the gene body (Fig 1C). Plotting the enrichment of major histone H2A variants and chromatin modifications over genes occupied by states E1-E8 supported the predominance of H2A.Z associated with H3K4me3 in the first nucleosomes while the gene body is occupied by H2A and H2A.X with H3K36me3 and H2Bub (Fig 1D). A succession of facultative heterochromatin states F3 − F5 occupied gene bodies (Fig 1C) although their organization was less complex compared with euchromatin states. Contrasting

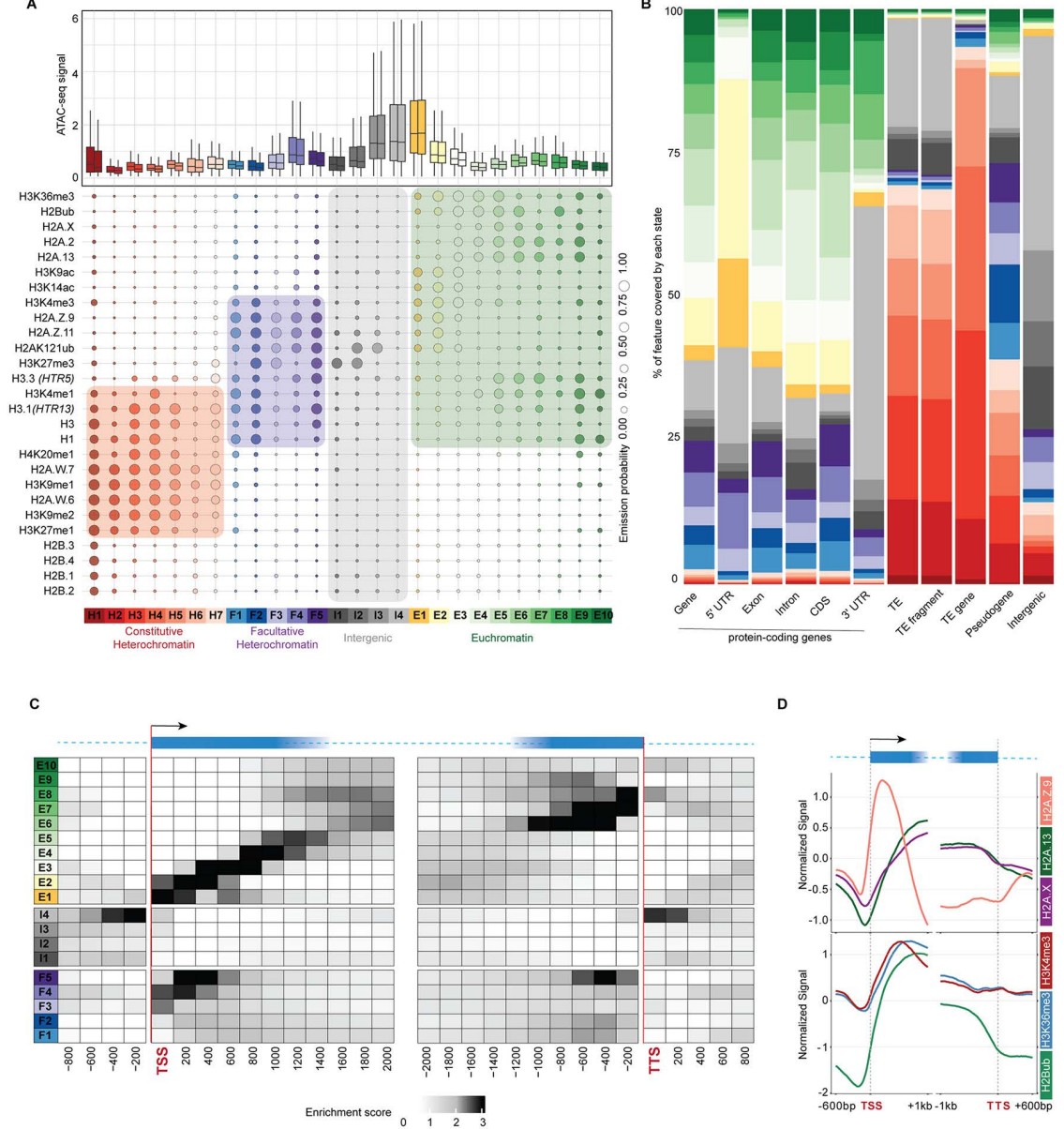

**Fig 1. Chromatin states of *Arabidopsis thaliana*.** (A) Bubble plot showing the emission probabilities for histone modifications/variants across the 26 chromatin states of *Arabidopsis thaliana*. The size of the bubble represents the emission probability ranging from 0 to 1. Coloured rectangles demarcate the classification of states into major domains of chromatin. The box plot on top shows the average ATAC-seq signal for each state representing chromatin accessibility. The two boxes per state are the two replicates of the ATAC-seq experiment. (B) Stacked bar plot showing the overlap between annotated genomic features and chromatin states. (C) Neighbourhood Enrichment analysis of chromatin states representing the fold enrichment for each state at fixed positions relative to the anchor position (TSS and TTS). (D) Metaplot showing the enrichment of prominent histone variants and modifications associated with active transcription. The enrichments here are plotted on genes overlapping states E1-E8. (n = 15234/ 27416 protein coding genes).

with the intergenic state I4, the other intergenic states I1, I2 and I3 were enriched in H3K27me3 and H2AK121ub, and showed lower gene expression (Figs 1A and S1A). These features suggest that the intergenic states I1, I2, and I3 are likely related to facultative heterochromatin, together occupying ca 30% of the genome (S1G Fig). Contrasting with the graded patterns of other chromatin states, the states E9 and E10 (ca. 5% of euchromatin) and the states F1 and F2 did

not show a strong positional enrichment relative to TSS or TTS (Fig 1C), but shared enrichment in linker histone H1 and were associated with genes expressed at very low levels (Figs 1A and S1A), suggesting a specific role of H1 in defining these chromatin states in addition to the major role of histone PTMs and H2A variants.

## Conservation of chromatin states in land plants

To investigate the conservation of chromatin states in land plants, we used the same approach as above to define chromatin states in *Marchantia polymorpha*. This model liverwort shares a similar genome organization with model species of hornworts and mosses that, together, form the monophyletic group of bryophytes, which diverged from vascular plants about 450 mya [26]. To the set of publicly available histone-modification enrichment profiles [27,28], we added profiles of histone H2A variants detected with specific antibodies for H2A.X.1, H2A.X.2, H2A.Z, and H2A.M.2, which are expressed in vegetative tissues of *Marchantia* [29]. Enrichment of chromatin features was mapped onto the newest telomere-to-telomere chromosome-level genome assembly v7.1 [30]. Using ChromHMM we described fifteen chromatin states (Fig 2A; see Methods) which were annotated based on their specific enrichment in different genomic features (Fig 2B).

Intergenic states covered approximately a quarter of the genome (S2A Fig). Chromatin states associated with specific levels of gene expression (S2B Fig), chromatin accessibility (Fig 2A), and DNA methylation (S2C–S2E Fig). Based on the enrichment of known constitutive heterochromatin marks, including H3K9me1/2 and H3K27me1, we ascribed six distinct states to constitutive heterochromatin (MpH1–MpH6) (Fig 2A) that covered 39% of the genome (S2A Fig). These states were also distinguished by the lowest accessibility, enrichment in the H2A variant H2A.M.2 (Fig 2A), the highest degree of DNA methylation (S2C–S2E Fig) and were primarily present on transposable elements, with a minor representation on protein-coding genes (Fig 2B). The hallmark of heterochromatin states is their enrichment in H2A.M.2 and H3K9me2, but they are divided into three groups (Fig 2A). MpH5 and MpH6 do not carry additional marks, except for methylated cytosines in MpH5, while states MpH3 and MpH4 are enriched in H3K9me1 and H3K27me1. Unlike heterochromatin states in *Arabidopsis*, *Marchantia* heterochromatin states MpH1 and MpH2, which represent more than a third of constitutive heterochromatin are also enriched in H3K27me3 with less DNA methylation than MpH3 and MpH4.

Facultative heterochromatin states MpF1 and MpF2, both enriched in H2A.Z and H2A ubiquitination (H2Aub), were associated with the lowest range of gene expression and covered ca 16% of the genome (Figs 2A, 2B, S2A, and S2B) and the state MpF1 was also enriched in H3K27me3. As in *Arabidopsis*, facultative heterochromatin states were also enriched in H3K4me1 and H3K4me3 in *Marchantia*. Neighborhood enrichment analysis anchored on TSS/TTS from MpTak-1v7.1 [30] revealed that intergenic state MpI1 was positioned upstream of MpF1. These two H3K27me3-enriched states together represented one third of facultative heterochromatin, exhibited low accessibility, and primarily marked repressed genes (Figs 2A–C, S2A, and S2B). States MpI2 and MpF2 occupied the remaining two thirds of facultative heterochromatin in *Marchantia*. Unlike MpF1, these states lacked enrichment of H3K27me3, associated with more accessible chromatin, and marked less repressed genes (Figs 2A–C and S2B), consistent with previous findings [31]. MpI2 occupied 19% of the genome and altogether facultative heterochromatin in *Marchantia* occupied 37% of the genome which was much higher than in *Arabidopsis*.

Regions with most open chromatin and the highest gene expression represented euchromatin (Figs 2A and S2B) and comprised the intergenic state MpI3 and four genic states E1–E4 covering altogether circa 24% of *Marchantia* genome (Figs 2A, 2B and S2A). The state MpI3 showed enrichment in the intergenic regions and the neighbourhood enrichment analyses anchored on TSS/TTS showed that the intergenic state MpI3 corresponds to a region located in intergenic regions upstream of the TSS (Fig 2B and 2C). Contrary to *Arabidopsis,* the promoters of *Marchantia* defined by the region just upstream of the TSS showed enrichment of H2Aub and the elongation mark H3K36me3, along with other euchromatic marks. We considered whether this could result from imprecise TSS definitions in the *Marchantia* genome. The current annotations are based on Iso-seq data [27], a long-read sequencing approach that accurately determines TSS positions, particularly for highly expressed genes. We further validated these annotations by examining CAGE-seq profiles, which

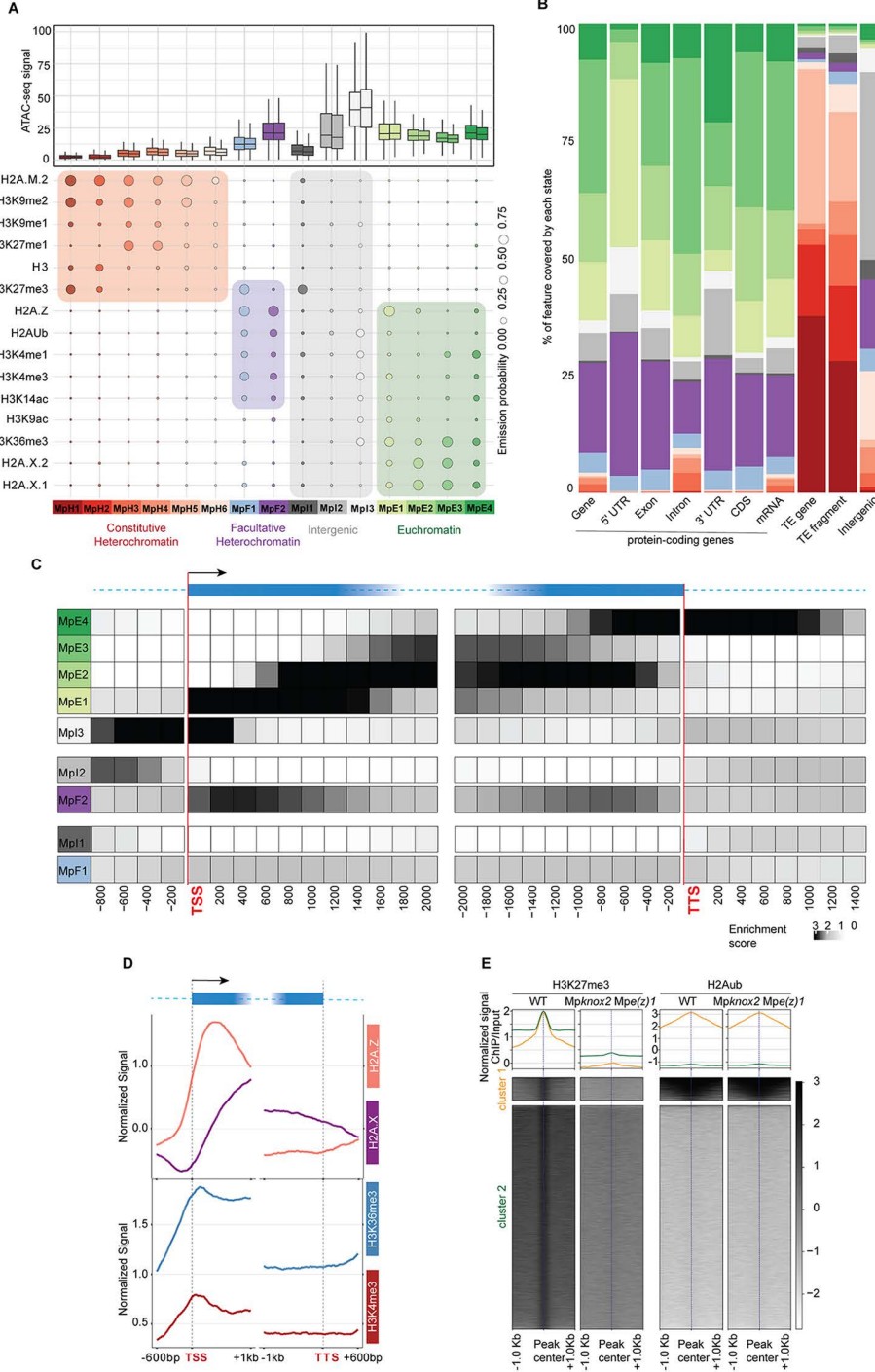

**Fig 2. Chromatin states of *Marchantia polymorpha*.** (A) Bubble plot showing the emission probabilities for histone modifications/variants across the 15 chromatin states of *Marchantia polymorpha*. The size of the bubble represents the emission probability ranging from 0 to 1. Coloured rectangles demarcate the classification of states into major domains of chromatin. The box plot on top shows the average ATAC-seq signal for each state representing chromatin accessibility. The two boxes per state are the two replicates of the ATAC-seq experiment. (B) Stacked bar plot showing the overlap between annotated genomic features and chromatin states. (C) Neighbourhood Enrichment analysis of chromatin states representing the fold enrichment for each state at fixed positions relative to the anchor position (TSS and TTS). (D) Metaplot showing the enrichment of prominent histone variants and modifications associated with active transcription. The enrichments here are plotted on genes overlapping states MpE1-MpE4. (n = 10715/ 17944 protein coding genes). (E) Enrichment heatmap and metaplots showing the apparent disassociation between H3K27me3 and H2AKub marks in wild

type and *Mpknox2 Mpe(z)1* double mutants. The enrichment of both marks is plotted on wild type H3K27me3 peaks (n = 25159). The peaks have been clustered using k-means clustering into two clusters where the cluster 1 represents peaks of H3K27me3 shared with H2AKub while cluster 2 represents majority of H3K27me3 peaks that do not overlap with H2AKub.

revealed that most CAGE-seq peaks [27,32] aligned with the annotated TSS positions (S2F Fig). While this supports the general accuracy of current TSS annotations, refinements in TSS definitions may help reconcile the observed enrichment patterns. It should also be noted that our chromatin state analysis uses 200 bp bins hence we cannot exclude the possibility that mapping of states on some genes may be offset by one bin in the upstream direction, an issue that future improvements in genome annotations may resolve. The most accessible, non-intergenic state is MpE1, which marked the + 1 nucleosome of most protein-coding genes of *Marchantia* (Fig 2A and 2C). This state, with strong enrichment of H2A.Z and H3K36me3, is present in the first few nucleosomes of active genes while the states MpE2–MpE4 successively occupy the gene body and are enriched in H2A.X (Fig 2A, 2C, and 2D). To evaluate the relative role of histone H2A variants and histone PTMs in defining chromatin states, we calculated a new matrix of chromatin states in the absence of data on either H2A variants or histone PTMs and used the Jaccard index to compare both resulting matrices to the original 15 chromatin states (S2G Fig). A high Jaccard index reports high similarity. We observed that *in silico* removal of H2A variants perturbed the definition of constitutive heterochromatin states less strongly than removal of histone PTMs while the influence of H2A variants and histone PTMs was comparable on the chromatin states of facultative heterochromatin and euchromatin. Accordingly, the pattern of enrichment of H2A.Z and H2A.X over expressed genes were similar between *Marchantia* and *Arabidopsis* (Figs 1D and 2D).

To directly compare the chromatin states of *Arabidopsis* and *Marchantia*, we first calculated a 15-states model in *Arabidopsis* based on analogous set of histone modifications and H2A variants used to identify chromatin states of *Marchantia*. The 15 and the 26 states models were highly comparable with a notable reduction of the complexity of constitutive heterochromatin to only four states and of euchromatin to only five states (Fig 1A and S3A). To compare the 15 chromatin states of *Arabidopsis* and *Marchantia*, we performed hierarchical clustering of emission probabilities of the two models (S2H Fig). The resulting tree supported the overall similarities of the constitutive heterochromatin and euchromatin states. In contrast, while the typical H3K27me3 enriched in facultative heterochromatin states of the two species clustered, the *Marchantia* facultative heterochromatin MpF2 clustered with the *Arabidopsis* euchromatic states E2, which was likely due to the prominent enrichment in H2A.Z in their definition. We noted that chromatin accessibility of MpI2 and MpF2 were similar to accessibility of euchromatin states than in *Arabidopsis* (Fig 2A and S3A) suggesting an ambivalent role for this subtype of facultative heterochromatin in *Marchantia*. To further compare the impact of PRC2 on histone modifications that define in a major manner the chromatin states, we analysed enrichment of H3K27me3, H2Aub, H3K4me3, H3K9me1, and H3K27me1 in the *Marchantia* Mp*knox2* Mp*e(z)1 double* mutant, which exhibited reduced PRC2 activity [28,33] (Fig 2E and S2I). As shown for PRC2 deficient mutants in *Arabidopsis* [34,35] peaks enriched in H3K27me3 in the wild type were depleted in the mutant deprived of PRC2 in *Marchantia*. However, the loss of PRC2 activity in *Marchantia* did not affect the deposition of H2Aub (Fig 2E) presumably by PRC1 activity [34]. Conversely, the loss of PRC1 affected the deposition of both H3K27me3 and H2Aub [34] suggesting a partial independence of the two Polycomb repressive pathways in *Marchantia*.

While our data supported an overall conservation of the chromatin states between the two species, we uncovered unique features of *Marchantia* chromatin organization. H3K27me3 was predominantly associated with transposable elements, with only a minor fraction marking protein-coding genes in facultative heterochromatin (Fig 2B). Instead, H2A.Z and H2AKub were the dominant marks of facultative heterochromatin, which is consistent with the mild effect of the loss of H3K27me3 on vegetative development [28] compared with *Arabidopsis* [22,36]. The promoter state MpE1 enriched in the transcription elongation mark H3K36me3, and the long persistence of E4 after the TTS creates an asymmetry between

the intergenic regions upstream and downstream of each gene (Fig 2C). This is in contrast with *Arabidopsis*, where the intergenic state I4 symmetrically occupies these two regions (Fig 2C). Notwithstanding potential inaccuracies in TSS/TTS coordinates, this asymmetry suggests that in *Marchantia*, gene orientation may define distinct chromatin environments in their vicinity, through mechanisms yet to be uncovered.

The distinct positional preferences of chromatin states in *Arabidopsis* versus *Marchantia*, particularly their sequential organization around gene regulatory regions, suggested a functional relationship with transcriptional regulation. To examine this in further detail, we ranked genes by their expression levels in *Arabidopsis* seedlings and *Marchantia* thalli. Highly expressed genes were depleted of facultative chromatin states in both species (Fig 3A and 3D).

In the case of *Arabidopsis*, I3 was also depleted in highly expressed genes and enriched in genes with low expression (Fig 3A and 3B). This observation was validated using experimentally derived TSS positions for *Arabidopsis* to examine the role of potential inaccuracies in TSS definitions (S4A–S4F Fig). The state E1, which was close to the TSS, was enriched in highly expressed genes in both species but more prominently in *Marchantia*, where it included the proximal promoter (Fig 3A and 3D).

Genes with low expression in vegetative tissues may be transcriptionally active at specific developmental stages or under environmental stimuli. To explore this, we assessed the coefficient of variation (CV) of gene expression across a panel of RNA-seq datasets spanning *Marchantia* and *Arabidopsis* development. The CV, defined as the standard deviation divided by the mean expression level, is independent of absolute expression and distinguishes ubiquitously expressed genes from those with variable expression in different conditions. Genes with high CVs exhibited greater coverage of facultative chromatin states, whereas ubiquitously expressed genes were largely depleted of these states (Fig 3B and 3E). In contrast, euchromatin states showed the opposite trend. A similar pattern was observed when analysing chromatin states within coding regions, with a much more prominent participation of euchromatin states (S4G–S4J Fig).

To determine whether facultative chromatin states are associated with specific tissues, we examined genes with tissue-specific expression in both species. Contrasting with ubiquitously expressed genes, promoters of these genes exhibited a strong association with facultative chromatin states (Fig 3C and 3F), supporting the hypothesis that facultative heterochromatin states are associated with cell differentiation and developmental transitions, with a prevalence of H3K27me3 enriched states. The loss of this marks resulted in strong ectopic expression of the genes covered primarily by both types of facultative heterochromatin (Fig 3G and 3H).

Besides differences related to the organisation of chromatin states on expressed genes and the relative extension of facultative heterochromatin marked by H3K27me3 versus H2Aub, our analyses conclude a broad conservation of chromatin states composition, and their association with transcriptional activity of genes and silencing of transposable elements in *Marchantia* and *Arabidopsis*.

## Transcription factors associate with specific chromatin states of *Arabidopsis*

The specific patterns of enrichment of histone modifications around the TSS both in expressed and repressed genes raised the possibility that specific transcription factors may preferentially associate with specific chromatin states. To investigate this hypothesis, we examined the genome-wide association patterns between transcription factors and chromatin states. To investigate TF binding preferences, we compiled a comprehensive dataset of ChIP-seq, DAP-seq, and position weight matrix (PWM)-based datasets from *Arabidopsis* and assigned TF-bound regions to chromatin states. TF occupancy combining data from ChIP-seq and DAP-seq was evaluated using an approach similar to that applied in human TF studies [5]. ChIP-seq peaks showed a strong preference for intergenic states (especially I3 and I4) and E1 (associated with the TSS) and were depleted in constitutive heterochromatin and euchromatin states present on bodies of expressed genes but interestingly some TF enrichment was found in states of facultative heterochromatin (Fig 4A), to a large extent consistent with the accessibility constraints (Fig 1A).

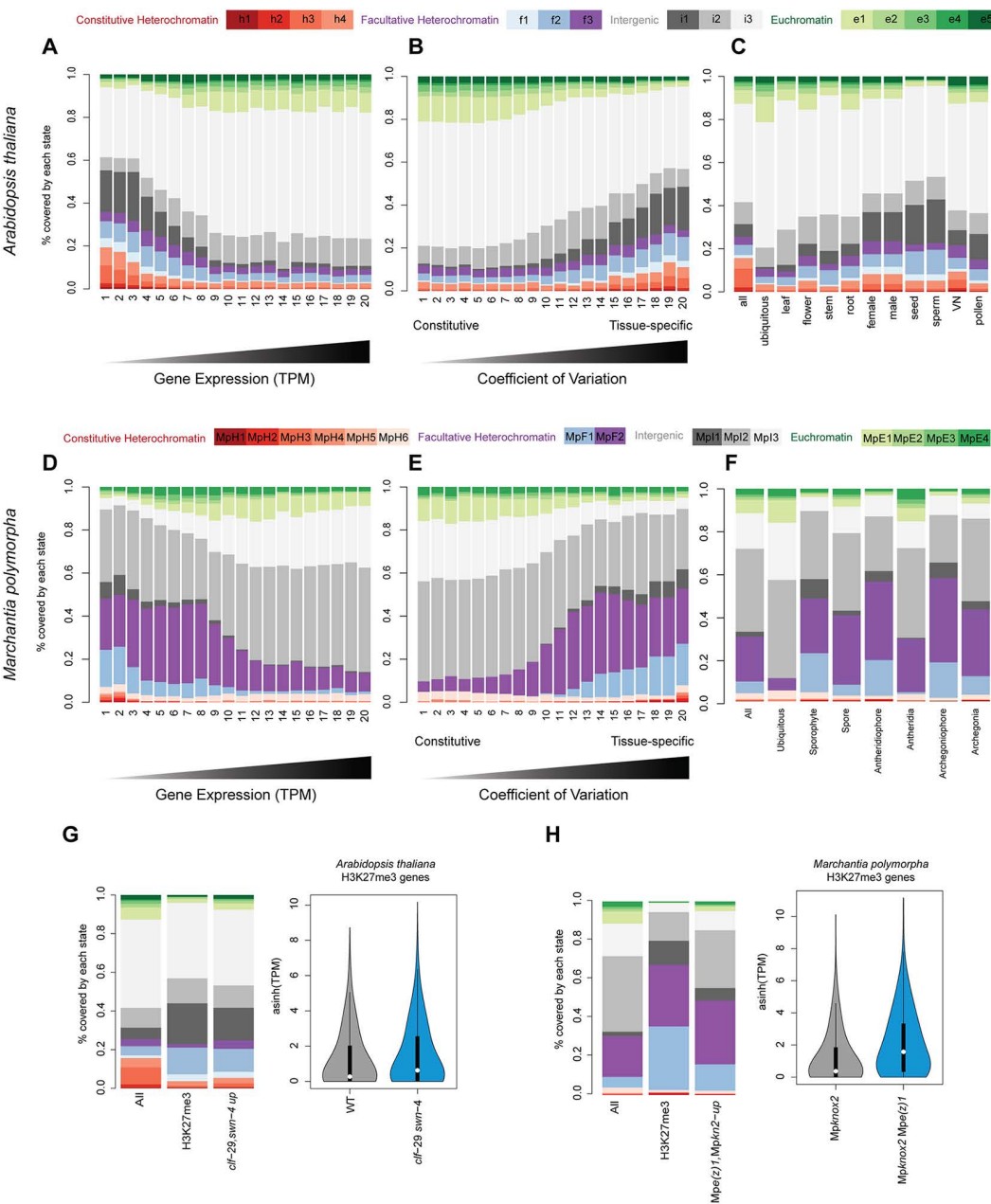

**Fig 3. Association between chromatin states and transcription across promoters in *Arabidopsis* and *Marchantia*.** (A, D) Bar plot of the proportion of chromatin states for 20 bins across promoters of genes ordered by transcripts per million (TPM) *Arabidopsis* leaves (A) and *Marchantia* thallus (D). (B, E) Bar plot of the proportion of chromatin states across promoters for 20 bins of genes ordered by coefficient of variation (CV) across different tissues of *Arabidopsis* (D) or *Marchantia* (E). (C, F) Bar plot the proportion of chromatin states across promoters for genes all genes or differentially expressed in specific tissues in *Arabidopsis* (C) and *Marchantia* (F). (G, H) Bar plot of the proportion of chromatin states across promoters for genes covered by H3K27me3 or differentially expressed in PRC2 mutant in *Arabidopsis* (G) and *Marchantia* (H). *clf/swn* double mutant for *Arabidopsis* and *Mpknox2 Mpe(z)1* double mutant for *Marchantia*, all genes are shown as a reference in both cases. In the right, asinh (TPM) of expression values for the same group of genes are shown for both mutant and the WT. The color code for the diagram represents the chromatin states in the 15 states models.

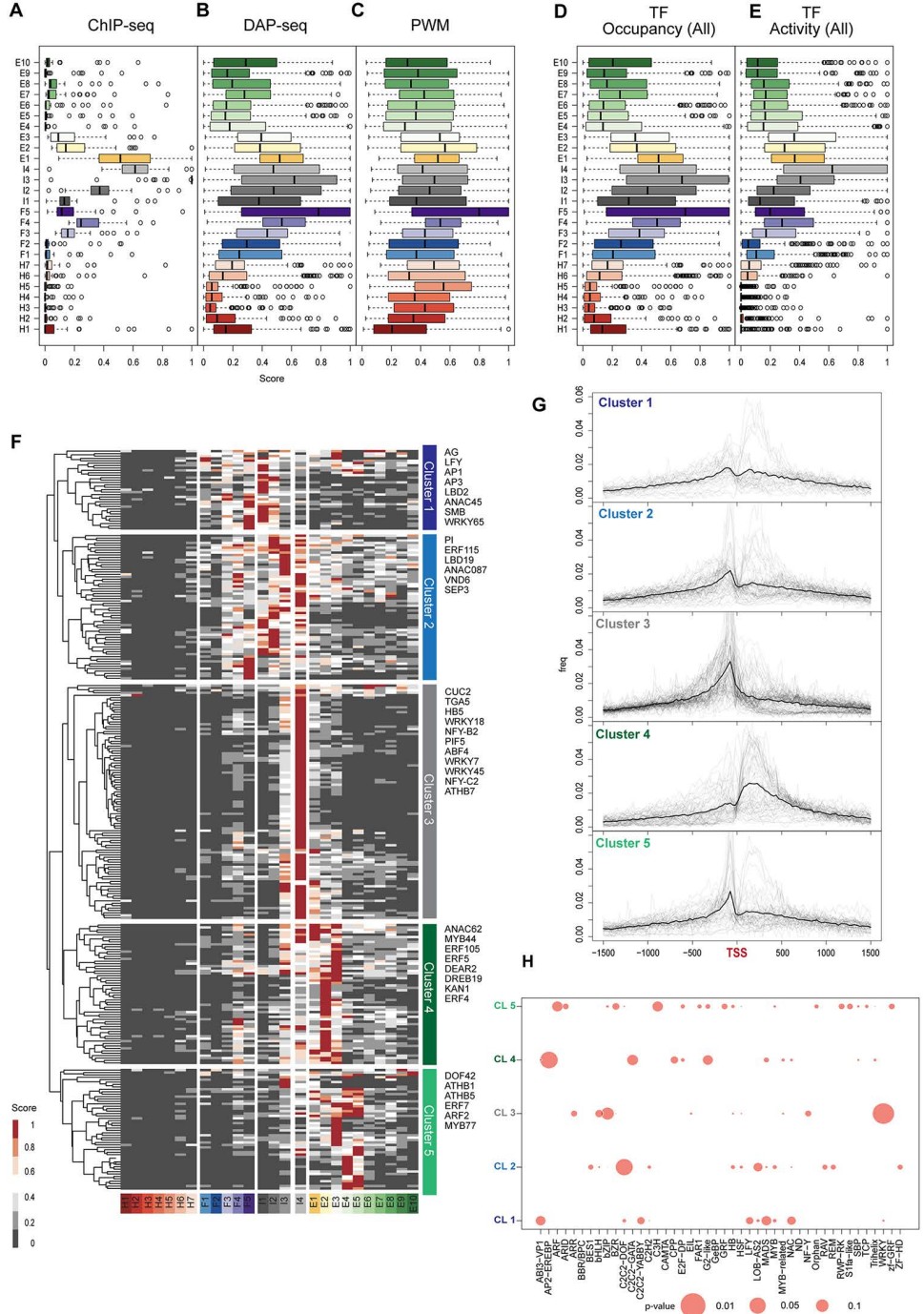

**Fig 4. Transcription factors activity shows heterogenous chromatin state preferences in *Arabidopsis*.** (A-C) Boxplot of TF occupancy scores based on ChIP-seq (A), DAP-seq (B), or position weight matrixes (PWM) motifs (C) in *Arabidopsis*. (D, E) Comparison of TF occupancy (D) and TF activity (E) scores across different chromatin states. The latter scores are calculated as the statistical enrichment of TF binding over putative targets (co-expressed genes). (F) Heatmap and hierarchical clustering of TF activity scores for ChIP-seq and DAP-seq experiments across chromatin states. Colour scale is shown at the bottom. Scores are normalized to the maximum value of each TF. Low unnormalized scores were discarded. The colour code for the diagrams represents the chromatin states in the current 15 states models. (G) Histogram of TF binding distribution relative to the transcription starting site (TSS) for *Arabidopsis* TFs from different clusters defined in Fig 4F. (H) Bubble plot of Fisher's Test of TF families across clusters defined in Fig 4F.

Prediction of TF binding sites using DAP-seq, which does not depend on chromatin accessibility [37], exhibited a broader distribution across chromatin states with some preference for intergenic states (Fig 4B). In contrast, PWM-based TF binding predictions, which only rely on DNA binding motifs, displayed a nearly uniform distribution (Fig 4C).

The observed differences between experimental approaches highlight the need for integrative analyses to fully resolve TF-chromatin state relationships in plants [5]. TF binding does not always lead to transcriptional regulation, as ChIP-seq or DAP-seq frequently detects peaks at sites that are not relevant for transcriptional activation or repression (Alvarez et al., 2021). To extend our analysis, we developed a method to estimate TF activity, incorporating co-expression data as a proxy for functional interactions [38]. For this, we classified the TF DNA-binding data between chromatin states and call a list of the proximal targets for each TF and chromatin states. We then extracted a list of all positively and negatively co-expressed genes for the TF that was used for that experiment. Finally, we calculated the enrichment between the two lists, for each state, against a random set of gene of equal size. This allowed us to distinguish between passive and active binding and reduce technical bias, particularly from *in vitro* binding. Ultimately, we hypothesize this score with highlight interactions between the chromatin and the TF that are functionally relevant. We adapted a method to calculate chromatin state preference scores by evaluating the enrichment of TF binding to their putative targets, positively and negatively co-expressed genes (see Methods). Overall, low scores of TF occupancy and TF activity were broadly correlated with in heterochromatin states, and the highest scores were observed in chromatin accessible regions such as the intergenic states (Fig 4D and 4E), which is the main factor that differentiated ChIP-seq from DAP-seq and PWM (Fig 4A–4E).

With this method to estimate TF activity, the scores of TF occupancy and activity converged between the different methods. To look at different patterns of chromatin preferences among TFs, we kept only the data from ChIP-seq and DAP-seq experiments and eliminated experiment with low scores across all chromatin states (score<8). In total, we collected binding profiles corresponding to ~300 TFs in *Arabidopsis*. Using hierarchical clustering, we identified distinct associations between TFs and specific chromatin states (Fig 4F). Interestingly, the different TF cluster also display distinct binding preferences relative to the TSS (Fig 4G). This observation was validated using experimentally derived TSS positions for *Arabidopsis* to examine the role of potential inaccuracies in TSS definitions (S5A–S5C Fig).

Unlike *Arabidopsis*, *Marchantia* lacks a comprehensive dataset of ChIP-seq or DAP-seq experiments and unfortunately our efforts to map the linker histone H1 and the variants of histone H3 and H2B have not been successful. However, previous studies have shown that TF DNA binding preferences are highly conserved across land plants [39], allowing to derive PWMs in *Marchantia* from PWMs from orthologous *Arabidopsis* proteins. Here, we combined recently published 24 DAP-seq experiments for *Marchantia* TFs with PWM from protein-binding microarrays [40] to serve as an alternative approach to infer TF binding. As with *Arabidopsis*, PWM-derived occupancy was evenly distributed across chromatin states (S2J Fig). However, when TF activity was estimated using co-expression, a striking shift emerged (S2K Fig). The most accessible chromatin states MpI2 and MpF2 were the most enriched, followed by MpE2, mirroring the patterns observed in *Arabidopsis*. A clustering analysis revealed further preferential association of clusters of TFs with states enriched either before the TSS (MpI1 and MpI2) or after the TSS (MpF2 and MpE2) (S2L Fig). However, the scarcity of the data and the fact that we compared TF binding in diploid tissues of *Arabidopsis* with haploid tissues in *Marchantia* limited our analysis and further investigation is required to determine whether similar regulatory strategies are employed in the interplay between TFs and chromatin states to establish the degree of conservation of these interactions across land plants.

Remarkably in *Arabidopsis,* many TF families showed preferential association with one or two chromatin states (Fig 4H) that may be associated with different mechanisms of action. The largest group (Cluster 3, 32% of all TFs) was associated primarily with I4 and bound upstream of the TSS (Fig 4F and 4G). This cluster comprised TFs from the WRKY, bHLH, bZIP, and HB families that bound to open chromatin of expressed genes (state I4) to regulate gene expression (Fig 4H and S3 Table), likely following the classic mode of RNA Polymerase II recruitment and stabilization of the pre-initiation complex [41]. A GO term analysis indicated that TFs of cluster 3 primarily participated in biosynthetic processes, photosynthesis, circadian rhythm, and other housekeeping functions. Other TFs associated with euchromatic chromatin

states formed clusters 4 and 5. Notably, TF families present only in cluster 5 were involved in hormone signalling and flower development, including ARF [42]. Most TFs from cluster 4 clearly bound after the TSS and several of these were associated with the control of the cell cycle, including E2FA, which is a typical marker of cell division. Clusters 4 and 5 also included the recently described GATC binding TFs [43], a large fraction of the AP2/EREBP family involved in stress response [44,45] and a large fraction of C2C2-GATA and G2-like TFs.

On the other hand, the TFs of Cluster 1 and 2 preferentially bound sites associated with facultative chromatin states (Figs 1A and 4F). Cluster 2 contained TFs primarily controlling development, and more particularly root and vascular development. Hence these TFs presumably control genes, which are not strongly active in leafy tissues of seedlings, in agreement with cluster 2 TFs occupying facultative heterochromatin states I1-I3 (Fig 4F). Many of these TFs binding sites were located upstream of the TSS (Fig 4G) and are thus expected to bind to typical promoters. In contrast, cluster 1 TFs primarily associated with the position of the first nucleosome downstream of the TSS (Fig 4G). Cluster 1 TFs are not expressed at the developmental stage analyzed (S5D Fig) and their loci are occupied by facultative heterochromatin. As a result, predicting the first nucleosome position on their target genes relies entirely on TSS annotation rather than chromatin state information, making these predictions less certain compared to the expressed TFs in clusters 3–5. Most of these TFs controlled meristem and flower development and were significantly enriched in MADS-domain TFs, including APETALA1, PISTILLATA, APETALA3, SEPALATA3, and the pioneer factor LEAFY (LFY), which regulates flower development. Together with their target genes, these TFs are not expressed in seedlings where the chromatin states were described [22,46] (Fig 4F and S5 Table), which aligns with their binding to facultative heterochromatin. Overall, our results demonstrate that chromatin states can serve as a framework to classify TF-chromatin interactions. This analysis revealed distinct regulatory strategies among plant TFs involving proximal promoters in intergenic regions, the +1 nucleosomes, or cis elements downstream of the TSS most likely acting as enhancers or silencers.

## Discussion

Here, we identified chromatin states in vegetative tissues of the liverwort *Marchantia* and found that their composition and association with transcription are strikingly conserved with the chromatin states of *Arabidopsis*. Overall, orthologous H2A variants and histone PTMs in *Marchantia* show the same associations and are equally important in defining chromatin states as has been shown in *Arabidopsis* [15]. The variants H2A and H2A.X and the PTM H3K36me3 mark euchromatin; H3K27me3 and H2A.Z mark facultative heterochromatin; in constitutive heterochromatin, H3K9me1/2 associates with the variants H2A.W or H2A.M, which share C-terminal motifs enriched in lysine residues [20]. Over protein-coding genes, chromatin states assemble in staggered arrays from the TSS to the TTS. Based on the integration of several datasets to map TF binding, we identified the preference of certain families and functional groups of TFs for specific chromatin states present either in the 5'UTR, the +1 nucleosome, or downstream of the TSS, suggesting a high complexity of cis elements controlling gene expression and the involvement of chromatin states in TF recruitment, either through direct recognition of chromatin marks or through co-regulatory mechanisms.

The evolutionary distance between the two species and their distinct genome organization makes *Marchantia* an ideal model for exploring the conservation of chromatin states and highlighting the evolution of fundamental chromatin regulatory mechanisms. Overall, the organization of chromatin states in *Marchantia* confirms our previous evaluation of profiles of chromatin modifications and their conservation amongst bryophytes [28]. Similarly, chromatin states described in *Arabidopsis* are representative of flowering plants [47,48]. We observed species-specific heterochromatin organization, including the dispersed constitutive heterochromatin, the less complex and less abundant facultative heterochromatin in *Marchantia* marked by H3K27me3, a predominant share of regions with this mark also associated with TE in bryophytes [49]. In *Marchantia*, euchromatin organization differs from *Arabidopsis*. The marks H3K4me1 and H3K36me3 reflecting transcriptional elongation and confined to the gene bodies in *Arabidopsis*, extend beyond the TTS in *Marchantia*, suggesting that signals for transcriptional termination differ between flowering plants and bryophytes. However, we cannot

exclude that this pattern may result from a combination of technical limitations including missing transcripts in the current MpTak1v7.1 genome annotation (e.g., lncRNAs) or potential annotation errors (e.g., incorrect gene ends or read-through transcripts). Overall, together with the similarities in the organization of histone modifications among bryophytes [28] and flowering plants [15,48] our data support the conservation of chromatin states and their relationship with gene expression in land plants. It is remarkable that this conservation is seen when comparing haploid gametophytic tissues of bryophytes and diploid sporophytic tissues of angiosperms, suggesting that chromatin organization in extant land plants reflects the organization of chromatin states in the haploid ancestors of land plants. The broad conservation of active chromatin states is also supported by a recent survey of chromatin states across eukaryotes while plants have evolved specific forms of repressive chromatin [50]. Specifically, we show a certain degree of independence between chromatin states governed by PRC1 and PRC2 in *Marchantia*, which contrasts with animals that connect the activities of the two modifiers through hierarchical recruitment models of "writing" and "reading" histone modifications H2AK121ub and H3K27me3 [51,52]. These differences support conclusions from phylogenetic analyses, which place the origins of the two independent types of Polycomb activities in the LECA, followed by divergence of their associations during evolution of different eukaryotic groups [53]. Our results and the differences in the phenotypes of the mutants in PRC1 and PRC2 in *Arabidopsis* [54] further support the conclusion that PRC1 and PRC2 have remained largely independent from each other in plants.

In animals, distinct forms of association with chromatin distinguish functional types of TFs, including those associated with facultative heterochromatin [4,6]. Comparable systematic studies were missing in plants. Here we integrate stable TF-chromatin interactions identified with ChIP-seq with transient, context-dependent interactions and show that plant TFs associate with preferred chromatin states. Many TFs appear to follow a "canonical" mode of action, binding to open chromatin and regulating transcription by facilitating the assembly or stabilization of RNA polymerase II and the initiation of transcription (S6A Fig). In *Arabidopsis*, this mechanism is predominant among TFs that regulate constitutive genes or modulate expression quantitatively. However, TF binding in *Arabidopsis* is not always associated with measurable transcriptional changes [24]. Notably, open chromatin is potentially occupied by multiple TFs, suggesting extensive competition for binding. This could be due to open chromatin, which is associated with highly expressed genes and permissive for TF binding. Such regions crowded in TF binding are also referred to as highly occupied target (HOT) regions, which have been identified before both in plants (19) and animals [55]. These binding events are yet functional and likely important for gene regulation (19) but they can also be associated with redundant or passive activity [19].

Genes that require tight spatiotemporal regulation—remaining off in most tissues and activating only under specific developmental or environmental conditions, are more likely controlled by TFs capable of acting within facultative chromatin states. These TFs may interact with chromatin remodelers to induce local chromatin accessibility or directly access DNA protected by nucleosomes, resembling the pioneer TF paradigm described in animals [56,57]. LFY functions as a pioneer TF by binding nucleosome-occupied sites at its targets APETALA1 and AGAMOUS, altering chromatin accessibility via SWI/SNF chromatin remodelers [20,58]. MADS-domain TFs, bZIP TFs, LEAFY COTYLEDON1 (LEC1), TERMINAL FLOWER 1, FLOWERING LOCUS T, and MONOPTEROUS, have been proposed as potential candidate pioneer factors [59]. Further experiments including careful analysis of TSS in relevant cell types and chromatin profiling will be required to investigate the potential mechanism of action of TFs of clusters 1 and 2 to qualify as pioneer factors. Thus, TF preferences for specific chromatin states provide a sequence-independent axis for classifying plant TFs and inferring their regulatory modes. Generalizing this observation, our work captures the chromatin preference signature of pioneer TFs and expands the list of potential candidate pioneer TFs in plants.

## Materials and methods

### Reagents and tools

Reagents and Tools are listed in Table 1.

**Table 1. Reagents and tools.**

| Reagent/Resource | Reference or Source | Identifier or Catalog Number |
|---|---|---|
| **Experimental Models** | | |
| *Marchantia polymorpha* Tak-1 | [60] | NA |
| *Arabisopsis thaliana* Col-0 | Nottingham Arabidopsis Stock Centre | NA |
| **Antibodies** | | |
| Rabbit polyclonal Anti-H2A.X.antibody | This paper | NA |
| Rabbit polyclonal Anti-H2A.X.2 antibody | This paper | NA |
| Rabbit polyclonal Anti-H2A.Z antibody | This paper | NA |
| Rabbit polyclonal Anti-H2A.M.2 antibody | This paper | NA |
| **Chemicals, Enzymes and other reagents** | | |
| cOmplete Protease Inhibitor Cocktail | Roche | Cat#11697498001 |
| Protein A/G beads | Thermo Fisher Scientific | NA |
| Proteinase K | Thermo Fisher Scientific | NA |
| RNase A | Thermo Fisher Scientific | NA |
| MinElute PCR purification kit | Qiagen | NA |
| Ovation Ultralow Library Systems V2 | Tecan, Männedorf, Switzerland | NA |
| **Software** | | |
| BEDTools suite | | v2.27.1 |
| FastQC | | v0.11.5 |
| Trim Galore | | v0.6.5 |
| Bowtie2 | | v2.4.1 |
| SAMtools | | v1.9 |
| Picard | | v2.22.8 |
| deepTools | | v3.1.2 |
| ChromHMM | | v1.23 |
| valr (R package) | | v0.6.6 |
| Tidyverse (R package) | | NA |
| GenomicRanges (R package) | | v1.56 |
| circlize (R package) | | NA |
| ComplexHeatmap (R package) | | NA |
| ChIPseeker (R package) | | NA |
| pheatmap (R package) | | NA |
| MEME Suite (FIMO) | | NA |
| PANTHER | | NA |
| **Other** | | |
| Covaris E220 High-Performance Focused Ultrasonicator | Covaris | E220 |
| Covaris milliTUBE | Covaris | NA |
| Illumina HiSeq v4 sequencer | Illumina | NA |

## Methods and protocols

**Profiling of *Marchantia polymorpha* H2A variants using ChIP-seq.** ChIP experiments were performed using a previously described protocol with some modifications [29]. Two-week-old thalli started from gemmae of *Marchantia polymorpha* Tak-1 wild type were collected and cross-linked using 1% formaldehyde in 1×PBS under vacuum on ice

for 10 min. The cross-linking reaction was quenched by adding 2 M glycine to achieve a final concentration of 0.125 M under vacuum on ice for 10 min. Excess solution was removed from cross-linked tissue by blotting with paper towels. Cross-linked tissue was then snap frozen in liquid nitrogen and ground to a fine powder using mortar and pestle. The powder was transferred into a 50-ml plastic tube and suspended in 40 ml of MP1 buffer (10 mM MES-KOH pH 5.3, 10 mM NaCl, 10 mM KCl, 0.4 M sucrose, 2% (w/v) Polyvinyl pyrrolidone (PVP 10), 10 mM $MgCl_2$, 10 mM 2-mercaptoethanol, 6 mM EGTA, 1×cOmplete protease inhibitor cocktail). Suspended samples were then filtered twice through one layer of Miracloth, once through a 40 µm nylon mesh, and twice through a 10 µm nylon mesh. Filtered samples were centrifuged at 3000×$g$ at 4°C for 10 min, and the supernatant was discarded. The pellet was washed using 15 ml of MP2 buffer (10 mM MES-KOH buffer pH 5.3, 10 mM NaCl, 10 mM KCl, 0.25 M sucrose, 10 mM $MgCl_2$, 10 mM 2-mercaptoethanol, 0.2% Triton-X 100, 1×cOmplete protease inhibitor cocktail) three times. The final pellet was then resuspended in 5 ml of MP3 buffer (10 mM MES-KOH pH 5.3, 10 mM NaCl, 10 mM KCl, 1.7 M sucrose, 2 mM $MgCl_2$, 10 mM 2-mercaptoethanol, 1×cOmplete protease inhibitor cocktail) and centrifuged at 16 000×$g$ at 4°C for 1 h. After centrifugation, the supernatant was discarded, and the pellet was resuspended in 900 µl of covaris buffer (0.1% SDS, 1 mM EDTA, 10 mM Tris–HCl pH 8.0, 1×cOmplete protease inhibitor cocktail). The resuspended pellet containing the chromatin fraction was fragmented using a Covaris E220 High-Performance Focused Ultrasonicator for 15 min at 4°C (duty factor, 5.0; peak incident power, 140.0; cycles per burst, 200) in a 1-ml Covaris milliTUBE. Sheared chromatin was centrifuged at 20 000×$g$ at 4°C for 10 min, and the supernatant was transferred into a new 5-ml tube and diluted by adding 2.7 ml of ChIP dilution buffer (0.01% SDS, 1.1% Triton-X-100, 1.2 mM EDTA, 16.7mM Tris-HCl pH 8.0, 167 mM NaCl). Diluted chromatin was cleared by incubating with proteinA/G beads (Thermo Fisher Scientific, Waltham, MA, USA) at 20 rpm rotating at 4°C for 1 h. Beads were removed by magnetic racks and precleared chromatin was separated into five tubes and incubated with 1 µg of specific antibodies for histone H2A variants at 20 rpm spinning at 4°C overnight. Chromatin bound by antibodies was collected by incubating with protein A/G beads for 3 h. The beads were collected by magnetic racks and washed twice with a low salt wash buffer (20 mM Tris–HCl pH 8.0, 150 mM NaCl, 2 mM EDTA, 1% Triton X-100 and 0.1% SDS), once with a high salt wash buffer (20 mM Tris–HCl pH 8.0, 500 mM NaCl, 2 mM EDTA, 1% Triton X-100 and 0.1% SDS), once with a LiCl wash buffer (10 mM Tris–HCl pH 8.0, 1 mM EDTA, 0.25 M LiCl, 1% IGEPAL CA-630 and 0.1% sodium deoxycholate), and twice with a TE buffer (10 mM Tris–HCl pH 8.0 and 1 mM EDTA). Immunoprecipitated DNA was eluted using 500 µl elution buffer (1% SDS and 0.1 M $NaHCO_3$) at 65°C for 15 min. To reverse cross-link, eluted DNA was mixed with 51 µl of reverse cross-link buffer (40 mM Tris–HCl pH 8.0, 0.2 M NaCl, 10 mM EDTA, 0.04 mg ml$^{-1}$ proteinase K; Thermo Fisher Scientific) and incubated at 45°C for 3 h and then at 65°C for 16 h. After cross-link reversal, DNA was treated with 10 µg of RNase A (Thermo Fisher Scientific), incubated at room temperature for 30 min and purified using the MinElute PCR purification kit (Qiagen). ChIP-seq library was generated from ChIPed DNAs using Ovation Ultralow Library Systems V2 (Tecan, Männedorf, Switzerland). The ChIP-seq libraries were sequenced on illumina Hiseq v4 to generate 50 bp single end reads.

**ChIP-seq data collection and processing.** The publicly available ChIP-seq datasets for *Arabidopsis* were downloaded from our previous study [15]. A detailed description of the sources of raw files can be found in the (S3 Table). The description of publicly available ChIP-seq datasets and the datasets generated for this study to calculate the chromatin states of *Marchantia* is also available in the (S4 Table).

Raw BAM files were converted to FASTQ files using the "bamtofastq" sub-command of the "BEDTools suite" v2.27.1 [61]. Sequencing quality of the raw files was evaluated using quality reports generated by FastQC v0.11.5 [62]. Reads were trimmed using "Trim Galore" v0.6.5 (DOI: 10.5281/zenodo.5127898). The trimmed reads were aligned to the reference genome: TAIR10 in case of *Arabidopsis* and MpTak-1 v7.1 in case of *Marchantia* using Bowtie2 v2.4.1 [63] and further processed using SAMtools v1.9 [64] and BEDTools v2.27.1 [61]. Duplicates were removed using Picard v2.22.8 (https://broadinstitute.github.io/picard/) to generate the aligned BAM files. The correlation between ChIP samples was evaluated using the multiBamSummary and plotCorrelation functions of deepTools v3.1.2 [65]. The bamCompare function

of deepTools v3.1.2 was used to normalize the ChIP signal with Input/H3 and generate BigWig files with normalized ChIP signal. The code for the above-mentioned analysis can be found here: https://github.com/Gregor-Mendel-Institute/vikas_states_tf_study_2025.git.

**Chromatin state calculations.** The aligned BAM files generated as explained above were used to calculate the chromatin states using the BinarizeBAM and LearnModel commands of ChromHMM v1.23 [2] with default parameters and models ranging from 2-50 states were learned. BAM files were binarized in 200 bp bins for model learning to characterize chromatin states using the multivariate Hidden Markov Model (HMM) to identify the combinatorial spatial patterns in the ChIP-seq signal of multiple chromatin marks. The models with 2–50 chromatin states were generated in each case. A 26-state model was chosen for the full *Arabidopsis* datasets based on two reasons: (1) As the raw data used to calculate states was the same as the one used in [15] with the main difference being the employment of stricter alignment parameters and filtering of previously included low confidence reads. These changes did not lead to extensive differences from the 26-state model in [15] (S1E Fig). A follow up on all models between 23–29 states revealed that models with less than 26 states showed loss of complexity (merging of states associated with distinct genomic features) and states with more than 26 states showed appearance of states that can no longer be associated with distinct genomic features. Hence, we decided to pick the 26-state model to be optimal for our analysis (S1 Data).

The chromatin states of *Marchantia* were calculated with the same pipeline. Given the comparatively smaller set of chromatin marks available in *Marchantia*, a smaller number of chromatin states was to be expected. A 15-state model was chosen for *Marchantia* (S2 Data) using the similar strategy as in [15]. To compare the chromatin state of *Arabidopsis* and *Marchantia*, we used only the chromatin marks that were available in both species and generated a less complex, comparable 15-state model for *Arabidopsis* (S3 Data).

**Annotation of chromatin states.** The BED files of the chromatin states generated by ChromHMM were imported into R v4.3.2 (https://www.R-project.org/). The chromatin state regions were overlapped with the genomic features defined in *Arabidopsis* genome TAIR10 (GFF3 file downloaded from here) and *Marchantia* genome MpTak-1 v7.1 (GFF3 file downloaded from here) using the "bed_intersect" function of R package valr v0.6.6 [66]. The results were plotted as stacked bar plots using the Tidyverse [67] package of R (Figs 1B, 2B, and S3B).

The NeighbourhoodEnrichment command of ChromHMM v1.23 [2] was used to calculate the enrichment of each state relative to the Transcription Start Site (TSS) or Transcription Termination Site (TTS) as anchors that were extracted from aforementioned GFF files. The resultant matrix was imported into R v4.3.2 (https://www.R-project.org/) and the heatmap was plotted using the Tidyverse [67] package of R (Figs 1C, 2C, and S3H).

To evaluate the genomic localization of states, especially the heterochromatic states (H1:H7) in *Arabidopsis*, histograms with binwidth of 50,000 bp were plotted on chromosome 1 using the Tidyverse [67] package of R showing the preferential localization of heterochromatic states near the centromere region (S1F Fig).

We identified the states of constitutive heterochromatin by their strong enrichment in the regions containing Transposable Elements (TE), TE genes and TE fragments (Figs 1B, 2B, and S3B) along with high emission probabilities of know heterochromatic marks including H2A.W, H3K9me1/2 among others (Figs 1A, 2A, and S3A). The distinction between the states of euchromatin and facultative heterochromatin was made by the high emission probabilities of marks associated with each type of chromatin state. The states of intergenic region showed little to no enrichment of any specific mark, representing Nucleosome Free Regions (NFR) and constituted about 20% of the genomes in both *Arabidopsis* and *Marchantia*. (S1G, S2A, and S3C Figs). We decided to annotate the states with the following nomenclature strategy:

(1) Euchromatin: The states of euchromatin were named with letter "E" in *Arabidopsis* 26-state model, with letter "e" in *Arabidopsis* 15-state model, and with letters "MpE" followed by a number in *Marchantia* 15-state model. The order of the numbers was assigned based on their enrichment relative to the TSS going from TSS to gene body (Figs 1C, 2C, and S3H).

(2) Constitutive Heterochromatin: The states of constitutive heterochromatin were named with letter "H" in *Arabidopsis* 26-state model, with letter "h" in *Arabidopsis* 15-state model, and with letters "MpH" in *Marchantia* 15-state model. The order of the numbers was assigned based on their localization on chromosomes going from centromere to pericentromere in case of *Arabidopsis* states (S1F Fig).

(3) Facultative Heterochromatin: The states of facultative heterochromatin were named with letter "F" in *Arabidopsis* 26-state model, with letter "f" in *Arabidopsis* 15-state model, and with letters "MpF" in *Marchantia* 15-state model.

(4) Intergenic states: The states of intergenic regions were named with letter "I" in *Arabidopsis* 26-state model, with letter "i" in *Arabidopsis* 15-state model, and with letters "MpI" in *Marchantia* 15-state model.

**Analysis of chromatin states.** The emission matrices generated by ChromHMM were imported into R v4.3.2 (https://www.R-project.org/). The state names and colours were reassigned using the strategy aforementioned, followed by plotting the emission matrices as bubble plots using the Tidyverse [67] package of R (Figs 1A, 2A, and S3A).

The percentage of genome covered by each state was calculated as follows: (Total number of base pairs covered by each state/ Genome size) * 100. The results were plotted as a pie chart using the Tidyverse [67] package of R (S1G, S2A, and S3C Figs).

We compared the chromatin states published in [15] with our 26-state model by importing the BED files for both models into R followed by overlapping the two matrices to create the transition matrix containing the occurrences of state transition between the two models. The data was plotted as an alluvial plot using the "circlize" [68] package of R. Similar analysis also led to the comparison of the *Arabidopsis* 26 and 15-state models (S1E and S3I Figs).

To show the similarities between the states of *Arabidopsis* 15-state model and the states from extensive 26-state model, after removing the additional marks from the 26-state model that were missing in the 15-state model.

To show the similar association of chromatin marks between the states of *Arabidopsis* and *Marchantia* 15-states models, hierarchical clustering of the two emission matrices was performed. H2A.W of *Arabidopsis* and H2A.M.2 of *Marchantia* were used interchangeably for comparison purposes. The two matrices were imported into R and merged for the common marks. The heatmap and clustering was performed using ComplexHeatmap [69] package of R (S2H Fig).

**Validation of Neighbourhood Enrichment analysis using the experimentally determined TSS and TTS of *Arabidopsis*.** We used TAIR10 rather than Araport11 as the reference genome because several studies [70,71] have demonstrated that TAIR10 gene borders are better defined for protein-coding genes, which are the focus of our analysis. To investigate the influence of potential inaccuracies in TAIR10-derived TSS/TTS positions, we incorporated experimentally determined gene borders obtained from full-length RNA-seq and high-throughput 5' and 3' tag sequencing data [71]. We mapped the experimentally determined TSS/TTS peaks from [71] to their nearest TAIR10-annotated protein-coding genes, successfully updating the TSS/TTS positions for 19,102 genes. The distribution of the updated TSS/TTS as compared to TAIR10 based TSS/TTS positions was plotted as density plot using the Tidyverse (33) package of R (S7A Fig).

To assess the impact of potential inaccuracies in the TAIR10 derived TSS/TTS positions on our neighbourhood enrichment analysis, we used experimentally validated TSS/TTS positions derived from [71] to evaluate the neighbourhood enrichment using three different TSS/TTS definition sets: (1) 19,102 genes with experimentally updated TSS/TTS positions based on [71]; (2) all 27,206 protein-coding genes, comprising the 19,102 genes with experimentally updated TSS/TTS and the remaining 8,104 genes with TAIR10-based TSS/TTS; and (3) all 27,206 protein-coding genes using only TAIR10-based TSS/TTS positions (S7B–S7D Figs). The Jaccard Index of Similarities between the three sets of matrices was calculated and plotted using the Tidyverse (33) package of R (S7E Fig).

**RNA-seq, ATAC-seq, and Bisulfite-seq data analysis.** The Bisulfite-seq and ATAC-seq data for wild type *Arabidopsis* seedlings was downloaded from GEO accession: GSE146948 [72] while the RNA-seq data was downloaded from [73]. All data was aligned to TAIR10 using bowtie2 v2.4.1 [63] and further processed using SAMtools v1.9 [64] and BEDTools v2.27.1 [61]. BigWig files for ATAC-seq and Bisulfite-seq and counts file for RNA-seq was imported into R. The data was

overlapped with states regions file using the valr v0.6.6 [66]. The results were plotted as box plots using the Tidyverse [67] package of R (Figs 1A, S1A–S1D, and S3D–S3G).

The Bisulfite-seq, ATAC-seq, and RNA-seq data for *Marchantia* was downloaded from [27] accession numbers SRA: PRJNA553138 and PRJDB8530. All data was aligned to MpTak-1 v7.1 genome and the boxplots were generated with the pipeline mentioned above (Figs 2A and S2B–S2E).

**Association of chromatin states with expression.** To assess the relationship between chromatin states and gene expression, we extracted promoter sequences from *Arabidopsis* thaliana (TAIR10) and *Marchantia* polymorpha (Tak1 v7.1), defining promoters as the 1,000 bp region upstream of the transcription start site (TSS) using the promoter function from the GenomicRanges package (v1.56) in R. For each gene in *Marchantia* genome, we determined the overlap of these promoter regions with annotated chromatin states using the findOverlaps and pintersect functions.

For gene expression data, we compiled a diverse set of publicly available RNA-seq experiments from MarpolBase expression [74] for *Marchantia* and EvoRepro for *Arabidopsis* [75]. To examine chromatin state association with transcriptional output, we ranked genes by their expression levels—measured in transcripts per million (TPM)—in the primary vegetative tissues where chromatin state data were derived (*Arabidopsis* leaf and *Marchantia* thallus). Genes were grouped into 20 equal-sized bins based on TPM values, and for each bin, we calculated the average proportion of each chromatin state per bin and visualized the distribution.

Similarly, to assess chromatin state dynamics across broader expression datasets, we analysed the same set of RNA-seq samples and estimated transcriptional variability by calculating the coefficient of variation (CV) for each gene, defined as the ratio of the standard deviation to the mean TPM across samples. Genes were ranked by CV, divided into bins, and plotted as described above.

For tissue-specific expression, we identified genes with at least a two-fold change in expression relative to their average expression across all tissues and assessed chromatin state distributions accordingly. A similar approach was applied to analyse chromatin state occupancy over gene bodies. To identify genes marked by H3K27me3, we applied the method described in [27] with data from [27] and [33].

**Transcription factor occupancy and association with chromatin states.** To analyse TF occupancy, we compiled publicly available *Arabidopsis* thaliana ChIP-seq and DAP-seq BED files from multiple studies [19,37,76]. For *Marchantia* polymorpha, we inferred TF binding sites by mapping position weight matrices (PWMs) from [37] to the *Marchantia* v7.1 genome using FIMO (MEME Suite) [77]. In both cases, each significant peak region in the genome was classified according to chromatin state using the OverlapsAny function from GenomicRanges package in R. For that purpose, only the centre of the peak was used to assign the chromatin state.

TF binding enrichment for each chromatin state was calculated following [6]. Specifically, for each TF and chromatin state (s), enrichment was determined using: $(a_s/b)/(c_s/d)$, where $a_s$ is the total number of bases in a peak call in s; b is the total number of bases in a peak call; $c_s$ is the total number of bases in s; d is the total number of bases of the chromatin state.

To calculate TF activity, we incorporated gene co-expression data. First, genes associated with TF peaks were annotated using the ChIPseeker package [78] with default parameters. Then, co-expression data from ATTEDII [79] (version Ath-r.c3-1) was used to filter TF target genes. Both positively (score < 2000) and negatively (score > max(score) - 1000) co-expressed genes were retained. The same enrichment formula described above was applied to this filtered subset. After this filter, we used the same formula than before.

To calculate the score of TF activity, we assessed statistical Enrichment of co-Expressed TF targets in chromatin states, we performed Fisher's exact tests (alternative hypothesis = greater). For each chromatin state, we compared TF targets with positively or negatively co-expressed genes as described above. Peaks were annotated as described above, and gene IDs were extracted for each chromatin state. As null distribution, we considered a random sample of genes from the whole genome using Table 2.

**Table 2. Data to score TF activity.**

| Count (genes with peaks + co expressed) | count (genes with peaks) |
|---|---|
| Count (co expressed) | 27205 for *Arabidopsis or* 18328 for *Marchantia* |

The TF-chromatin state score was computed as: $-\log_{10}$(p-value) $+ \log_2$(odds ratio $+ 1$) and normalized using the maximum score for each TF.

For each method we generated an $N_{TF} \times N_{state}$ matrix containing all computed scores and visualized chromatin state occupancy patterns using the pheatmap R package. For aggregate analyses across all TFs, scores were summed and normalized to the maximum score for each state.

In subsequent analyses, TFs with maximum scores below 8 were filtered out to exclude low-confidence results. TF family enrichment was assessed using Fisher's exact tests, employing TF classifications from [37] and PlantTFDB [80]. GO term enrichment analysis of biological processes in *Arabidopsis* was performed using PANTHER [81] with default parameters.

## Supporting information

**S1 Fig. Genome-wide chromatin state profiling and functional characterization in *Arabidopsis thaliana*.** (A) Box plot showing the expression of protein-coding genes overlapping with each chromatin state in Transcripts per Million (TPM). (B-D) Box plot showing levels of CpG (B), CHH (C), and CHG (D) methylation for all chromatin states. (E) Flow diagram showing the overlap (in bp) between the chromatin states defined in this study with states from Jamge et al., 2023. The color code for the flow diagram represents the chromatin states in the current model. (F) Genomic distribution of heterochromatic states (H1–H7) from bottom to top on Chr1 of *Arabidopsis thaliana*. (G) Pie chart showing the percentage of the genome covered by each state.
(TIF)

**S2 Fig. Genome-wide chromatin state profiling and functional characterization in *Marchantia polymorpha*.** (A) Pie chart showing the percentage of the genome covered by each state of *Marchantia polymorpha*. (B) Box plot showing the expression of protein-coding genes overlapping with each chromatin state of *Marchantia polymorpha* in Transcripts per Million (TPM). (C-E) Box plot showing levels of CpG (C), CHH (D), and CHG (E) methylation for all chromatin states of *Marchantia polymorpha*. (F) Enrichment heatmap of CAGE-seq signal around the TSS defined in *Marchantia* genome MpTak1v7.1. The green rectangle represents the bin size reference for the chromatin state as calculated by chromHMM. (G) Heatmap showing the Jaccard similarity index between the states generated using the whole model and states using a subset of marks, i.e., excluding a set of marks and variants as indicated on the x-axis. (H) Hierarchical clustering of the emission probabilities of chromatin states of *Marchantia* and a down-sampled 15-states model of *Arabidopsis* representing the conservation of chromatin state definition in the two distantly related species. (I) Heatmap and metaplots showing the enrichment of H3K27me1, H3K9me1, and H3K4me3 marks in wild type and *Mpknox2 Mpe(z)1* double mutants. The enrichment of both marks is plotted on wild type H3K27me3 peaks (n = 25159). The two clusters are the same as obtained in Fig 2E. (J, K) Boxplot of TF occupancy (J) and TF activity (K) scores across different chromatin states. The later scores are calculated as the statistical enrichment of TF binding over putative targets (co-expressed genes). (L) Heatmap and hierarchical clustering of TF activity scores for PWM across chromatin states. Colour scale is shown at the bottom. Scores are normalized to the maximum value of each TF. Low unnormalized scores were discarded. The colour code for the diagrams represents the chromatin states in the current 15–states models.
(TIF)

**S3 Fig. Functional characterization of a 15-state chromatin model in *Arabidopsis thaliana*.** (A) Bubble plot showing the emission probabilities for histone modifications/variants for a down-sampled 15-state model of Arabidopsis thaliana.

The size of the bubble represents the emission probability ranging from 0 to 1. Coloured rectangles demarcate the classification of states into major domains of chromatin. The box plot on top shows the average ATAC-seq signal for each state representing chromatin accessibility. The two boxes per state are two replicates of the ATAC-seq experiment. (B) Stacked bar plot showing the overlap between annotated genomic features and the chromatin states in the 15-state model of *Arabidopsis thaliana*. (C) Pie chart showing the percentage of the genome covered by each state of the 15-state model of *Arabidopsis thaliana*. (D) Box plot showing the expression of protein-coding genes overlapping with each chromatin state of the 15-state model of *Arabidopsis thaliana* in Transcripts per Million (TPM). (E-G) Box plot showing levels of CpG, CHH, and CHG methylation respectively, for all chromatin states of the 15-state model of *Arabidopsis thaliana*. (H) Neighbourhood Enrichment analysis of chromatin states of *Arabidopsis* 15-state model representing the fold enrichment for each state at fixed positions relative to the anchor position (TSS and TTS). (I) Flow diagram showing the overlap (in bp) between the chromatin states of extensive 26-state model and the 15-state model *Arabidopsis thaliana*. The colour code for the flow diagram represents the colours for the states of the 15-state model.
(TIF)

**S4 Fig. Chromatin state dynamics across gene bodies in *Arabidopsis* and *Marchantia*.** (A, B) Bar plot of the proportion of chromatin states for 20 bins across coding regions of 19102 protein-coding genes with experimentally derived TSS and TTS from Ivanov et al., 2021 ordered by transcripts per million (TPM) (A) and coefficient of variation (CV) (B) in *Arabidopsis* leaves. (C, D) Bar plot of the proportion of chromatin states for 20 bins across coding regions of 19102 protein-coding genes with experimentally derived TSS and TTS from Ivanov et al., 2021 and the 8104 protein-coding genes with TAIR10 derived TSS and TTS ordered by transcripts per million (TPM) (C) and coefficient of variation (CV) (D) in *Arabidopsis* leaves. (E, F) Bar plot of the proportion of chromatin states for 20 bins across coding regions of all 27206 protein-coding genes with TAIR10 derived TSS and TTS ordered by transcripts per million (TPM) (E) and coefficient of variation (CV) (F) in *Arabidopsis* leaves. (G, H) Bar plot of the proportion of chromatin states for 20 bins across coding regions of genes ordered by transcripts per million (TPM) (G) and coefficient of variation (CV) (H) *Arabidopsis* leaves. (I, J) Bar plot of the proportion of chromatin states for 20 bins across coding regions of genes ordered by transcripts per million (TPM) (I) and coefficient of variation (CV) (J) *Marchantia* thallus.
(TIF)

**S5 Fig. TF binding distribution at TSS and developmental expression patterns in *Arabidopsis*.** (A) Histogram of TF binding distribution relative to the experimentally derived transcription starting site (TSS) of 19102 protein-coding genes (Ivanov et al., 2021) of *Arabidopsis thaliana* TFs from different clusters defined in Fig 4F. (B) Histogram of TF binding distribution relative to the experimentally derived transcription starting site (TSS) of 19102 protein-coding genes (Ivanov et al., 2021) and TAIR10 derived TSS for the remaining 8104 protein-coding genes of *Arabidopsis thaliana* TFs from different clusters defined in Fig 4F. (C) Histogram of TF binding distribution relative to the TAIR10 derived transcription starting site (TSS) of all 27206 protein-coding genes of *Arabidopsis thaliana* TFs from different clusters defined in Fig 4F. (D) Line plots showing the median expression levels of the TFs belonging to the 5 clusters shown in Fig 4F across the broad developmental stages of *Arabidopsis thaliana* development.
(TIF)

**S6 Fig. Model of transcription factor interplay with facultative and open chromatin to control transcription rates.** The histone icon was adapted from Bioicons: https://bioicons.com/icons/cc-by-4.0/Nucleic_acids/DBCLS/histone.svg. Acknowledgement: histone icon by DBCLS https://togotv.dbcls.jp/en/pics.html is licensed under CC-BY 4.0 Unported https://creativecommons.org/licenses/by/4.0/.
(TIF)

**S7 Fig. Impact of TSS/TTS annotation sources on chromatin state enrichment patterns in *Arabidopsis*.** (A) Strand-aware density of TSS and TES shifts (bp) between the Ivanov et al. (2021) calls and TAIR10 annotations. Kernel density

estimates (y-axis: probability density, 1/bp) show the distribution of per-gene shifts (in bp) between Ivanov et al. (2021) calls and TAIR10 protein-coding gene annotations for *Arabidopsis thaliana*. (B) Neighbourhood enrichment analysis of the chromatin states on 19102 protein-coding genes with experimentally derived TSS and TTS from Ivanov et al., 2021. (C) Neighbourhood enrichment analysis of the chromatin states on 19102 genes with experimentally derived TSS and TTS from Ivanov et al., 2021 and the 8104 protein-coding genes with TAIR10 derived TSS and TTS. (D) Neighbourhood enrichment analysis of the chromatin states on all 27206 protein-coding genes with TAIR10 derived TSS and TTS. (E) Jaccard similarity index estimating the similarity in TSS and TTS positions between the 19102 updated coordinates from Ivanov et al., 2021 and 19102 + (8104 coordinates from TAIR10) compared with TAIR10 TSS/TTS coordinates. The higher the jaccard index, the higher similarity between the two coordinates.
(TIF)

**S1 Table. Datasets used to define *Arabidopsis* chromatin states model.**
(XLSX)

**S2 Table. Datasets used to define *Marchantia* chromatin states model.**
(XLSX)

**S3 Table. List of *Arabidopsis* Transcription Factors enriched in each of the 5 clusters and related metadata.**
(XLSX)

**S4 Table. List of Transcription Factors studied in *Marchantia* and related metadata.**
(XLSX)

**S5 Table. (A–E) Gene Ontology lists for cluster 1–5 of *Arabidopsis.***
(XLSX)

**S1 Data. BED files containing the chromatin states segments data files of *Arabidopsis* 26 and 15–state models and *Marchantia* 15–state model.**
(BED)

**S2 Data. MP 5 segments.**
(BED)

**S3 Data. AT 15 segments.**
(BED)

## Acknowledgments

We thank the entire Berger group, for their insightful, considerate, and helpful discussions, as well as Matt Watson for editorial assistance during the preparation of the manuscript. We thank the Molecular Biology Service and Media Kitchen for a constant supply of plates, basic reagents, and cloning and sequencing services, as well as the Peptide Synthesis Service. Additionally, we thank the Vienna BioCenter Core Facilities, in particular the Next Generation Sequencing facility for their advice and swift handling of all our requests and helpful discussions. For open access purposes, the authors have applied a CC BY public copyright license to any author accepted manuscript version arising from this submission.

## Author contributions

**Conceptualization:** Frederic Berger, Facundo Romani.

**Data curation:** Vikas Shukla, Elin Axelsson, Jim Haseloff, Facundo Romani.

**Formal analysis:** Vikas Shukla, Elin Axelsson, Facundo Romani.

**Funding acquisition:** Tetsuya Hisanaga, Frederic Berger, Facundo Romani.

**Investigation:** Vikas Shukla, Elin Axelsson, Tetsuya Hisanaga, Facundo Romani.

**Methodology:** Vikas Shukla, Elin Axelsson, Tetsuya Hisanaga, Facundo Romani.

**Project administration:** Frederic Berger, Facundo Romani.

**Resources:** Vikas Shukla, Elin Axelsson, Tetsuya Hisanaga, Facundo Romani.

**Software:** Vikas Shukla, Elin Axelsson, Facundo Romani.

**Supervision:** Frederic Berger.

**Validation:** Vikas Shukla, Elin Axelsson, Facundo Romani.

**Visualization:** Vikas Shukla, Facundo Romani.

**Writing – original draft:** Vikas Shukla, Frederic Berger, Facundo Romani.

**Writing – review & editing:** Vikas Shukla, Elin Axelsson, Frederic Berger, Facundo Romani.

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
