## [Decision Letter · Decision Letter 0]

25 Dec 2025

Dear Dr Berger,

We are pleased to inform you that your manuscript entitled "Chromatin State Architecture governs Transcription Factor Accessibility across Plant Genomes" has been editorially accepted for publication in PLOS Genetics. Congratulations! Unexpectedly both reviews came back in time to send out this positive response on December 25, so we hope this also provides some festive cheer for your co-authors.

Yours sincerely,

John M. Greally, D.Med., Ph.D.

Section Editor

PLOS Genetics

John Greally

Section Editor

PLOS Genetics

Aimée Dudley

Editor-in-Chief

PLOS Genetics

Anne Goriely

Editor-in-Chief

PLOS Genetics

BlueSky: @plos.bsky.social

Comments from the reviewers (if applicable):

Reviewer's Responses to Questions

**Comments to the Authors:**

Reviewer #1: I was original reviewer #1 for ReviewCommons.

The authors addressed my concerns. The authors added new figures and text that improved/clarified key parts. I see no reservations that justify a delay of this publication from my end.

It is an impressive analysis, I hope it will be useful the community.

Reviewer #2: The authors addressed all of my previous concerns. I have no further requests.

**Have all data underlying the figures and results presented in the manuscript been provided?**

Reviewer #1: Yes

Reviewer #2: Yes

PLOS authors have the option to publish the peer review history of their article (what does this mean? ). If published, this will include your full peer review and any attached files.

**Do you want your identity to be public for this peer review?** For information about this choice, including consent withdrawal, please see our Privacy Policy .

Reviewer #1: No

Reviewer #2: No

**Data Deposition**

http://datadryad.org/submit?journalID=pgenetics&manu=PGENETICS-D-25-01313

**Press Queries**

---

## [Editor Report · Acceptance letter]

PGENETICS-D-25-01313

Chromatin State Architecture governs Transcription Factor Accessibility across Plant Genomes

Dear Dr Berger,

We are pleased to inform you that your manuscript entitled "Chromatin State Architecture governs Transcription Factor Accessibility across Plant Genomes" has been formally accepted for publication in PLOS Genetics! Your manuscript is now with our production department and you will be notified of the publication date in due course.

With kind regards,

Lilla Horvath

PLOS Genetics

On behalf of:
